# Potential for Ammonia Generation and Emission in Broiler Production Facilities in Brazil

**DOI:** 10.3390/ani13040675

**Published:** 2023-02-15

**Authors:** Fernanda Campos de Sousa, Ilda de Fátima Ferreira Tinôco, Vasco Fitas Cruz, Matteo Barbari, Jairo Alexander Osorio Saraz, Alex Lopes da Silva, Diogo José de Rezende Coelho, Fatima Baptista

**Affiliations:** 1Department of Agricultural Engineering, Federal University of Viçosa, Viçosa 36570-900, Brazil; 2Departamento de Engenharia Rural, Escola de Ciências e Tecnologia, MED—Instituto Mediterrâneo para a Agricultura, Ambiente e Desenvolvimento, Universidade de Évora, Évora 7000-849, Portugal; 3Department of Agriculture, Food, Environment and Forestry (GESAAF), Università degli Studi di Firenze, 13-50145 Firenze, Italy; 4Department of Agricultural Engineering, Universidad Nacional de Colombia, Medellin 050034, Colombia; 5Department of Animal Science, Federal University of Viçosa, Viçosa 36570-900, Brazil

**Keywords:** air quality, animal buildings, environmental modelling, gas emissions, sustainability

## Abstract

**Simple Summary:**

The aim of the present study was to develop predictive models for the potential generation and emission of ammonia from Brazilian broiler chicken production. For this study, samples of poultry litter (shavings and coffee husks) from thirty commercial poultry houses located in the Zona da Mata Mineira region were analyzed. The poultry litter samples were subjected to different air temperatures in climatic chambers. The models developed and validated showed high accuracy, indicating that they can be used to estimate the potential for generation and emission of ammonia in poultry production, enabling its quantification when its measurement is not possible.

**Abstract:**

Air quality is one of the main factors that must be guaranteed in animal production. However, the measurement of pollutants is still a problem in several countries because the available methods are costly and do not always apply to the reality of the constructive typology adopted, as in countries with a hot climate, which adopt predominantly open facilities. Thus, the objective of the present study was to develop predictive models for the potential generation and emission of ammonia in the production of broiler chickens with different types of litter, different reuse cycles and under different climatic conditions. Samples of poultry litter from thirty commercial aviaries submitted to different air temperatures were analyzed. The experiment was conducted and analyzed in a completely randomized design, following a factorial scheme. Models were developed to predict the potential for generation and emission of ammonia, which can be applied in facilities with ambient conditions of air temperature between 25 and 40 °C and with wood shaving bed with up to four reuse cycles and coffee husks bed with up to six reuse cycles. The developed and validated models showed high accuracy indicating that they can be used to estimate the potential for ammonia generation and emission.

## 1. Introduction

Brazil is currently the world’s largest exporter of chicken meat and the third-largest chicken meat producer in the world, with more than 14 million metric tons in 2021 [1]. For more than a decade, the country has maintained the position of the world’s largest exporter of this type of meat, with shipments to more than 150 countries [2]. Brazilian broiler production employs more than 3.5 million workers, directly and indirectly, accounting for almost 1.5% of the Gross Domestic Product (GDP), representing social and economic relevance [3].

However, the confinement of animals in the facilities can generate impacts on the environment and ecosystems [4,5,6,7,8,9]. Therefore, the Brazilian national animal production industry must comply with environmental control programs to reduce the impact of its activities [10,11,12,13,14,15]. In this respect, many studies have repeatedly demonstrated that, among the environmental impacts, air quality is one of the main factors to be guaranteed by the animal production [16,17,18,19,20,21]. However, the presence of atmospheric pollutants generated in animal production facilities, such as: ammonia (NH_3_), particulate matter (PM), hydrogen sulfide (H_2_S), nitrous oxide (N_2_O), methane (CH_4_), carbon dioxide (CO_2_), and volatile organic compounds (VOCs) can compromise air quality [4,22,23,24]. Ammonia in poultry facilities is mainly produced from the decomposition of uric acid present in animal manure [7,25]. In this uric acid degradation process, ammonia and carbon dioxide are generated [4,25,26], According to the Intergovernmental Panel on Climate Change the ammonia is considered as a primary indirect source of N_2_O [4,27]. Thus, although ammonia is not a greenhouse gas, it indirectly contributes to the formation of important greenhouse gases such as nitrous oxide and carbon dioxide [27].

In addition, ammonia stands out among the pollutants present in the facilities because it is present in higher concentrations in aviaries compared to other pollutants [28,29,30,31,32]. And also because it generates economic and financial losses in production, as it favors susceptibility to diseases in animals and workers [4,23,32,33,34,35]. High levels of ammonia (above 25 ppm) in facilities affect animal productivity due to damage caused to body weight gain, feed conversion, welfare, carcass quality and even the immune system of chickens. In addition to irritation of the conjunctiva, cornea and mucous membranes of the respiratory tract increasing susceptibility to respiratory infections [4,32,34,35,36]. All of these issues regarding local ammonia problems in the facility depend on the concentration of ammonia and the time of exposure to ammonia [37,38,39,40]. The concentration of ammonia in the facilities is related to the potential for ammonia generation, which represents the maximum amount of ammonia that can be generated.

In this sense, the ammonia emission in the aviaries is caused by several factors related to the avian bed. As pH high and, mainly, the excess moisture from the waste [41]. When the ammonium ion (NH_4_^+^), present in said compound, it is converted to ammonia (NH_3_) it diffuses from the bed to the environment to be a volatile compound [18]. In addition, the ammonium ion (NH_4_^+^), through nitrification and denitrification processes, can be converted to nitrous oxide (N_2_O), an important greenhouse gas [42].

Another very important aspect to consider is that, as far as our knowledge, there isn’t an indicator relating the potential for ammonia generation and emissions with variables that are easier to measure and/or obtain in animal production facilities such as characteristics of the thermal environment (air temperature) and facility management (bed type and bed reuse cycle) making very difficult the establishment of emission standards or indices for a specific installation. In addition, the quantification of air pollutants present in animal production facilities is expensive and still unfeasible in many countries with significant animal production [17,18,19,35].

There are a variety of methods to quantify the concentration of ammonia in installations, ranging from simpler and less expensive equipment to sophisticated and expensive methodologies [18,36,39,43,44,45,46,47,48]. Among the main ones used, we can mention: electrochemical sensors [35,49,50,51,52,53], denuders [44,45,54], photoacoustic spectrophotometry [46,55,56], chemiluminescent analyzers [43,45].

Among the methodologies available to quantify ammonia emission in animal production facilities, one can cite the calculation of the ammonia emission rate established as a product of concentration and ventilation rate [18]. However, if on the one hand the determination of the ammonia concentration can be considered simple, the determination of the ventilation rate represents a challenge [17,19,31,57,58], mainly in conditions of complex air flow and/or without the possibility of exact definition of the air inlets and outlets in the installation. Tracer gas or carbon dioxide mass balance methods also present challenges due to the lack of control over the volume of air in the facility [8,31,53,57,59,60,61]. The use of passive flow samplers and flow chambers are well used at the research level, but their use is compromised in the field given the difficulty of obtaining certain chemical reagents and laboratory processes, in addition to not allowing a real-time response [35,50,51,52,62,63,64]. Another way to quantify ammonia is through dispersion modeling, which also depends on wind direction.

Not to mention that the methodologies available for quantification of ammonia don’t always apply to the reality of the constructive typology of the facilities. As for example in countries with a hot climate that adopt predominantly open facilities, without side walls in the installations, using only a screen, which makes the internal environment subject to external variations [65,66], in addition to making it impossible to control the volume of internal air and making it difficult to define air inlets and outlets in the installation [17,18,67,68]. The prediction of the potential of ammonia generation and emission from broilers production, will be an important advance since it becomes possible to evaluate the pollutant level of a particular poultry house, thus serving as the base for fair fines or taxes incentives.

Also, these models are essential to build reliable inventories of generation of ammonia and emission control strategies, which are still poor in Brazil and in South America. The prediction models of potential generation and emission of ammonia can also contribute to improve the accuracy and simplicity of the predictions, thus reducing the need for measurements, which are usually complex and costly [69]. Therefore, the main objective of this work was to develop predictive models for ammonia generation and emission potentials in the Brazilian broilers production.

## 2. Materials and Methods

This study is part of the PhD thesis of Sousa (2018) [70]. The animal study protocol was approved by the Ethics Committee in Animal CEUA/UFV (protocol 09/2015 23 April 2015) in agreement with the actual Brazilian legislation (Lei N° 11.794, 2008), Normative Resolutions edited by CONCEA/MCTI, the DBCA (Brazilian Practice Guideline for the Care and Use of Animals for Scientific Purpose and Teaching) and the Guideline of Practice the Euthanisia recommended by CONCEA/MCTI.

### 2.1. Collection of Poultry Manure Samples in Broiler Production Facilities

Samples of different types of poultry manure with different reuse cycles were collected in facilities to produce broiler chickens in the state of Minas Gerais (MG). All facilities were subjected to the same climatic conditions because they were in the same region, in the Zona da Mata of Minas Gerais (Figure 1). According to the Köppen classification, the climate in this region is of the Cwb type–tropical climate of altitude, with rainy summers and mild temperatures.

The facilities used (Figure 2) had the following constructive characteristics: orientation of the ridge east-west, width between 12 and 14 m, height of 2.8 to 3.2 m, length between 100 and 140 m, sides with walls of 20 cm and closing with screens and mobile plastic curtains. The breeding system consisted of a bed over the floor, with predominantly male animals of the Cobb lineage, in a housing density between 14 and 18 birds/m^2^. The facilities remained predominantly open and were equipped with a positive pressure ventilation system, with axial fans arranged on the sides of the sheds that were triggered automatically according to the setpoint for the temperature control.

In order to obtain homogeneous, representative samples and with the largest quantity of wastes in the bed, the collections of the bed samples were made during the last week of creation, that is, between 35 and 42 days of life of the animals. We evaluated aviary beds constituted by two types of substrates, wood shavings and coffee husks. The wood shaving beds were reused for up to four cycles and the coffee husk beds for up to six cycles.

For each of the different bed types and reuse cycles, samples were collected in three facilities, constituting three replicates. In each one, samples were collected in 20 different predetermined points, distributed throughout the entire length of the facility. The collection points were representative of the whole area of housing and areas near or on feeding and watering systems (lines) were avoided [72]. The samples were taken with a shovel [73]. From these samples, for each facility a single composite sample was formed, decompressed, homogenized and stored in a suitably identified plastic bag. After the collection, the composite sample was immediately sent to the climatic chambers where the experiment was developed.

### 2.2. Conduction of the Experiment in Climatic Chambers

The experiment was carried out in 4 climatic chambers (Figure 3a) built in the dimensions of 2.4 m × 3.3 m × 2.5 m (height, length and width), located in the experimental area of the Center for Research in Ambience and Engineering of Agroindustrial Systems (AMBIAGRO), from the Department of Agricultural Engineering (DEA) of the Federal University of Viçosa (UFV), in Viçosa, Minas Gerais. Each climate chamber was equipped (Figure 3b) with an air conditioning apparatus type hot/cold 12,000 BTU/h, an electrical resistance heater with a power of 2000 W and a humidifier air with mist flow rate of 300 mL/h with a capacity of 4.5 L. The conditioner, heater and humidifier were activated or turned off by electronic digital control of temperature and humidity (MT-531Ri plus, Full Gauge Controls, Canoas, Brazil), according to pre-established values. The air renewal inside the climatic chambers was done by means of axial exhausts, activated automatically.

The aviary bed samples were individually arranged in rectangular plastic boxes (Figure 4) with a volume of 20 L in the following dimensions: 60 cm long, 38 cm wide and 16 cm high. According to the depth adopted in the facilities for broiler production the boxes were filled with the aviary bed up to the height of 10 cm. In the center of the box it was placed an equipment for fixation of the volatilized ammonia (Figure 4), according to the methodology SMDAE [74]. This method, called “Saraz Method for Determination of Ammonia Emissions” (SMDAE), is based on the mass diffusion method for the determination of ammonia flux from broiler bedding, based on the total volatilized ammonia content that is volatilized and captured. The NH_3_ capture device uses a PVC tube 20 cm in diameter and 30 cm high and two polyurethane sponges 20 cm in diameter and 2 cm thickness positioned inside the tube at 10 cm (sponge 1) and 30 cm (sponge 2) from the base of the PVC collector. Sponge 1 captures the flow of ammonia by the bed, and sponge 2 prevents contamination of the sample by external gases [74].

During a 24-h period, four air temperatures were tested: 25, 30, 35 and 40 °C, in each climate chamber. In all climatic chambers, the same range of relative humidity was adopted, between 40 and 70%, considered ideal for animal thermal comfort [65,66]. The air renewal rate was the same in all climatic chambers, using the minimum ventilation, with 6 air renewable per hour [75]. Thus, all the environmental factors agreed with the most commonly practiced in the Brazilian industry of broilers production. To evaluate the physical and chemical properties of the avian bed, two-bed samples were collected for laboratory analysis, one immediately before starting the experiment, called the initial condition, and another after the 24-h period of exposure to different treatments.

The moisture content of the bed was determined using the method recommended by the Brazilian Research and Agricultural Company [76]. The pH of the bed samples was determined by the potentiometric method [77].

The total nitrogen of the samples was determined by the Kjeldahl method [78] following the digestion, distillation and titration processes. To determine the ammoniacal nitrogen, the Kjeldahl method was also used [78], as suggested by APHA [79] the distillation and titration processes of the samples were carried out.

The ammonia generation potential was determined based on the ammoniacal nitrogen content (Equation (1)), since this is constituted by free ammonia (NH_3_) and ammonium ions (NH_4_^+^) that can be converted to ammonia depending on air and bed variables, such as temperature, humidity and pH [69].
[NH_3_-N]_l_ = 1000 × [TAN]/{K_f_ × 10^−pH^/K_d0_ + MC × (1+ 10^−pH^/K_d0_)/ρ_H2O_} (1)
where: [NH_3_-N]_l_ is the concentration of dissolved phase NH_3_-N (µg L^−1^); [TAN] is the total ammoniacal nitrogen content in bed on a dry basis (µg g^−1^); K_f_ is the Freundlich partition coefficient (L kg^−1^); pH is the hydrogen potential; K_d0_ is the dissociation constant in water (dimensionless); MC is the moisture content of bed on a dry basis (*w*/*w* %) and ρ_H2O_ is the density of water (kg L^−1^).

The ammonia emission potential was developed based on the ammonia emission flux calculated using the SMDAE method [74]. The SMDAE mass flux was obtained by using the Equation (2).
(2)$$\mathrm{SMDAE}\;\left({\mathrm{mgNH}}_{3\;}m^{-2}s^{-1}\right)=\frac{{\mathrm{NH}}_{3\;}}{\mathrm{At}}$$
where: SMDAE is NH_3_ mass flux (g NH_3_ m^−2^ s^−1^); NH_3_ is NH_3_ mass (g NH_3_); A is sponge area (m^2^) and t is exposure time of sponge (s).

### 2.3. Experimental Design and Statistical Analysis

The experiment was conducted and analysed according to a completely randomized design, following a 4 × 4 factorial scheme for the wood shaving bed (4 cycles of bed reuse and 4 different temperatures, 25, 30, 35 and 40 °C); and a 6 × 4 factorial scheme for the coffee husk bed (6 cycles of bed reuse and 4 different temperatures, 25, 30, 35 and 40 °C). The analyses were conducted separately for the beds of wood shavings and coffee husk, due to the different number of reuse cycles obtained for each type of bed.

First, the resulting data were submitted to a variance analysis (ANOVA) to verify the interaction effect between the cycles of use and the air temperatures used. The analysis was conducted using the SAS MIXED procedure [80], and adopted a 0.05 significance level.

In order to estimate the ammonia generation and emission potential, multiple equations were adjusted as a function of the reuse cycle and temperature, and the linear, quadratic and cubic effects of these variables were tested. Subsequently, the models were reduced by excluding non-significant variables using the “backward” method. All equations were adjusted using the SAS procedure REG [80], considering a 0.05 significance level.

To assess the adequacy of multiple equations developed to predict potential generation and emission of ammonia, a cross-validation analysis was performed [81]. The analysis was performed from 1000 simulations, using the least squares linear function of R [82] and the packages “boot” and “mass”. For each simulation, the original database was randomly split into two new subsets of approximately the same size. The first subset (generation subset) was used to obtain the estimates and the second subset (test subset) was used to test the estimates to obtain the statistical adequacy measures. After each simulation, the database was reorganized, and the whole process repeated 1000 times, to take the average of adequacy statistics.

These results were used to estimate the accuracy and precision of the developed equations using the mean square of prediction error (QMEP), correlation coefficient and concordance coefficient (CCC) and coefficient of determination (R^2^). The QMEP was decomposed into three main sources of variation: (1) mean deviation, which represents the central tendency of the deviation; (2) systematic deviation, which represents the slope deviation of 1; and (3) random error, which represents variations that are not explained by regression [83].

The CCC was used to simultaneously access the accuracy and precision of the equations and was decomposed into an estimated correlation coefficient (ρ) which estimates the accuracy of the model and the deviation correction factor (C_b_) that indicates the accuracy of the equations. The values of CCC, ρ and C_b_ ranged from 0 to 1, where values close to 1 indicate precise/accurate equations [83,84]

## 3. Results and Discussion

According to the analysis of variance, no interaction effect (*p* > 0.05) was observed between cycle of use and temperature on any of the analyzed variables (pH, humidity, total nitrogen and ammoniacal nitrogen). Indicating that the variables presented similar pattern as a function of the cycles of use, for all tested temperatures.

### 3.1. Physical and Chemical Characterization

#### 3.1.1. Moisture Content

In general, the initial moisture values of the wood shaving bed and coffee husk bed were higher than the moisture values of the beds after being subjected to different thermal conditions for all reuse cycles tested (Figure 5). The values of moisture in the shaving bed in the first cycle of use ranged from 32 to 36%, showing a drop to about 28% in the fourth cycle of use (Figure 5a). The bed of coffee husks showed a decreasing behavior of moisture, in the linear case, due to the increase in the number of cycles of use, and in the first cycle of use an average value of 30% was observed, while in the sixth cycle of use an average content of 25% was obtained (Figure 5b).

The moisture content of the wood shaving bed and coffee husk bed showed a significant difference between the cycles of use (*p* < 0.05). For the shaving bed, both the initial values and the values after the beds were submitted to the treatments presented quadratic behavior decreasing due to the increase in the number of cycles of use. Likewise, the moisture content of the coffee husk bed presented a decreasing behavior, in the linear case, due to the increase in the number of cycles of use (Table 1).

For both types of bed evaluated, the moisture content decreased as temperatures (treatments) increased. Such behavior has been previously observed by several authors [18,41,85], evidencing that the temperature of the air influences the humidity of the bed, that is, the higher the temperature of the drier air will be the bed.

Among the techniques adopted in avian facilities with bed reuse it is common to incorporate a small substrate layer at the beginning of each new productive cycle. The incorporation of this new substrate can cause a reduction in the moisture content of the bed, since, according to data obtained in this study, the pure wood has moisture content in the range of 7% and the coffee husk around 15%, values well below beds, even after the first use cycle. Another technique of handling the avian bed commonly used to control indoor air quality is the application of chemical additives, such as the application of agricultural gypsum, which promotes the reduction of moisture content and reduces the volatilization of ammonia [18,86,87].

The fall in the moisture content of the bed with the increase in the number of cycles of use and with the increase of the temperature, described by the equations presented above, can also occur due to the loss of water that occurs in the residues and in the bed by evaporation processes. As the installations remain predominantly open, their interior is subject to the occurrence of air currents that promote the circulation of air inside the installation contributing to these evaporative processes.

#### 3.1.2. pH

Initial pH values (before beds were submitted to treatments) ranged from 8.5 to 8.2 for shaving bed, and from 8.7 to 8.5 for coffee husk (Figure 6). Similar values, ranging from 7.9 to 8.5, were observed by Marín et al. (2015) [41], for both shaving and coffee husks beds, from the first to the fourth reuse cycle.

The pH values varied in a decreasing quadratic manner, according to the increase in the number of cycles of use, for both types of bed (Table 2). After the beds are submitted to the different thermal treatments, an increase in the pH values is observed in relation to the values of the initial condition, thus evidencing that the air temperature can influence in the ammonia emission indirectly by promoting increase in the pH values.

These high pH values show the existence of ammonia generation and emission potential. For several authors [85,88,89] high pH values contribute to high ammonia emission rates. The higher the pH of the wastes, the higher the relation between NH_3_ and NH_4_, and in the range between 8.0 and 10.0 this maximum ratio [88], and the optimum pH for the degradation of uric acid is around 9.0 [90]. Considering these two factors together, the maximum potential of ammonia generation and emission must occur when the waste has pH around 9.0, because both the highest conversion rate of NH_4_ in NH_3_ and the higher bacterial activity of decomposition of uric acid, since, in this pH range, optimal conditions exist for the development of the decomposing bacteria.

The decreasing quadratic behavior according to the increase in the number of cycles of use may be related to the behavior of the moisture content, which also decreased with the cycles. As can also be related to the same reasons mentioned above, since the addition of a new bed layer can change the pH, because, according to analyzes carried out in this study, the pure wood has pH in the range of 4.0 and coffee husk in the band of 5.0. In addition, the application of chemical additives, such as aluminum sulphate, promotes the reduction of pH values, thus reducing the bacterial load and consequently decreasing the volatilization of ammonia [18,86,87].

#### 3.1.3. Total Nitrogen

The total nitrogen content varied on average between 20 and 38 g·kg^−1^ for wood shaving bed and between 35 and 45 g·kg^−1^ for coffee husk bed, from the first to the fourth cycle (Figure 7). Unlike the results found in the present study, Marin et al. (2015) [41] found lower total nitrogen contents, average values from the first to the fourth cycles, of 16.3 g·kg^−1^ for wood shavings and 19.3 g·kg^−1^ for coffee husk bed.

The total nitrogen contents presented, for both beds, a quadratic behavior increasing according to the bed cycle, regardless of the temperature used (Table 3). The increase in the total nitrogen content with the increase in the number of cycles of bed utilization occurs due to the increase of nitrogen in the bed during the breeding cycles due to the waste load that is deposited in each new use of the bed.

According to the experimental conditions of this study, when the beds were submitted to the different thermal treatments, only losses by volatilization of ammonia occurred, without any form of nitrogen increment. Consequently, it can be observed that the higher total nitrogen contents were observed at lower temperatures, being this behavior more outstanding for the bed of shavings. Thus higher air temperatures influence the losses of nitrogen to the environment by favoring the volatilization of ammonia, as reported by several authors [85,88].

#### 3.1.4. Ammonia Nitrogen

The concentration of ammoniacal nitrogen, regardless of the tested temperature, varied between 7 and 9 g·kg^−1^ for both wood shavings and coffee husks, from the first to the last cycle of use analyzed (Figure 8). Lower values than those found in the present study were observed by Marin et al. (2015) in fine shavings and coffee husk beds, between the first and fourth cycles of use, found values of ammoniacal nitrogen between 1 and 2.5 g·kg^−1^.

The concentration of ammonia nitrogen increased in a quadratic manner according to the cycles of use for the two bed types, regardless of the temperature tested (Table 4). In the present study, for both litter, shavings and coffee husks, the values of ammoniacal nitrogen corresponded to about 25% of the total nitrogen, this result is close to the value of 30% of the total nitrogen that is volatilized in the form of ammoniacal nitrogen, according to Atia (2008) [91].

The highest values of ammoniacal nitrogen, both for shavings and coffee husks, were observed in the treatment of 40 °C, again showing that the elevated temperature contributes to the higher ammonia generation potential.

For both types of beds there is a more significant increase in ammonia nitrogen contents up to the third cycle of use and a stabilization trend from the third cycle. This increase in the ammoniacal nitrogen more evident in the first cycles of use can be explained by the behavior of the data referring to the values of pH and humidity of the bed, which presented a similar behavior, being higher in the first cycles, which favored both the decomposition of the uric acid, leading to the formation of NH_3_, as well as the actual conversion of NH_4_ into NH_3_. All these factors together have collaborated with the greatest potential for ammonia generation in bed. These results corroborate Marín et al. (2015) [41], which, when evaluating coffee bark beds and fine shavings with up to four cycles of use, concluded that the ammonia nitrogen values of the avian beds tend to increase with each reuse of the bed.

### 3.2. Potential for Ammonia Generation

The potential of ammonia generation for wood shavings and coffee husks beds with different reuse cycles and submitted to different air temperatures could be determined by means of the multiple prediction equations of ammonia generation (Equations (3) and (4)). The “backward” process of selection of variables showed as significant only for linear and quadratic effect for the reuse cycle and linear for temperature. Additionally, no intercept significance was observed for any of the developed models (*p* > 0.05).
Generation of NH_3_ (wood shavings) = 0.028 × T + 6.301 × C − 1.059 × C^2^(3)
Generation of NH_3_ (coffee husks) = 0.046 × T + 4.247 × C − 0.512 × C^2^(4)
where: Generation of NH_3_ (g·kg^−1^); T = air temperature (°C); C = reuse cycles.

Cross-validation indicated a high accuracy for both models of ammonia generation through C_b_ values of 0.970 and 0.978 for wood shavings and coffee husk, respectively. In addition, the QMEP partition indicated low prediction errors associated with the fixed variables since most of the prediction errors were associated with random errors of 63.76 and 82.53%, for wood shavings and coffee husks, respectively. The equations presented moderate and low accuracy with R^2^ values of 0.559 and 0.265 and ρ values of 0.747 and 0.514, which produced moderate CCC values of 0.722 and 0.497 for wood shavings and coffee husk, respectively (Table 5).

The model for prediction of the potential of ammonia generation for wood shavings bed presented high accuracy and moderate to low accuracy (Figure 9a). While the prediction model of the ammonia emission potential for coffee husk bed presents high accuracy and moderate accuracy (Figure 9b). However, the high accuracy demonstrated by both models indicates that the equations are suitable for use in estimating ammonia generation, since accuracy is the most important measure for a prediction model because it represents the model’s ability to predict values real [83].

### 3.3. Ammonia Emission Potential

The potential emission of ammonia from aviary beds, wood shavings and coffee husks, under different air temperatures was determined according to the multiple equations to predict the emission of ammonia as a function of air temperature and the bed reuse cycle (Equations (5) and (6)). Significance was observed for the linear and quadratic effects of the cycle of use and linear for temperature, according to the “backward” process of selection of variables. No significant effect of the intercept was observed (*p* > 0.05) for any of the developed models.
Emission of NH_3_ (wood shavings) = 0.00031 × T − 0.00280 × C + 0.00086 × C^2^(5)
Emission of NH_3_ (coffee husks) = 0.00024 × T − 0.00014 × C + 0.00023 × C^2^(6)
where the emission of NH_3_ is expressed in g·m^−1^·s^−1^.

Cross-validation indicate a high accuracy for both models of ammonia emission prediction through C_b_ values of 0.970 and 0.997 for wood shavings bed and coffee husks bed, respectively. In addition, the QMEP partition indicated low prediction errors associated with the fixed variables since most of the prediction errors were associated with random errors of 91.47% and 96.54%, for shaving and coffee husk, respectively. The equations presented moderate to high accuracy with R^2^ values of 0.792 and 0.861 and ρ values of 0.889 and 0.927, which produced high CCC values of 0.883 and 0.924 for shaving and coffee husk, respectively (Table 6).

The prediction model for the emission of ammonia to wood shavings bed showed high accuracy and moderate precision (Figure 10a), on the other hand, the prediction model developed for coffee husk bed showed high accuracy and high precision (Figure 10b).

The high accuracy demonstrated by the equations to predict the emission of ammonia, for both wood shavings bed and coffee husks bed, indicates that such equations are appropriate, as mentioned before [83].

## 4. Conclusions

For both types of aviary beds analysed (wood shavings and coffee husks), moisture contents and pH values decreased as a function of the increase in cycle of use, regardless of the temperature of the air tested. While the values of ammonia and total nitrogen presented an increasing behaviour due to the increase in the number of cycles of reuse of the avian bed in the different air temperatures evaluated. Therefore, the increase in the number of cycles of use of avian beds contributes to the increase of ammonia emissions in the production facilities of broilers, especially when conditions of air temperatures above 25 °C inside these facilities.

The potential for ammonia generation and emission can be estimated using the models developed for production facilities of broilers subjected to air temperatures between 25 to 40 °C, with wood shavings bed reused for up to 4 cycles or coffee husks bed reused for up to 6 cycles. The developed models to predict the ammonia generation and emission in facilities of broilers present as predictive variables the number of cycles of reuse of the bed and the air temperature. Since they presented high accuracy, verified through the cross-validation technique these models are suitable for estimating the ammonia generation and emission potential, under the conditions studied. However, such models are limited to the type of bed substrate used in the present study (wood shavings and coffee husks), the number of bed reuse cycles (up to 4 cycles for wood shavings and up to 6 cycles for coffee husks), the air temperature of the study (25, 30, 35 and 40 °C) and the same condition of constant minimum ventilation in bed. Therefore, there is still a need for more research to enable the generation of more robust models to estimate the potential for ammonia generation and emission in facilities under real conditions of animal production.

## Figures and Tables

**Figure 1 animals-13-00675-f001:**
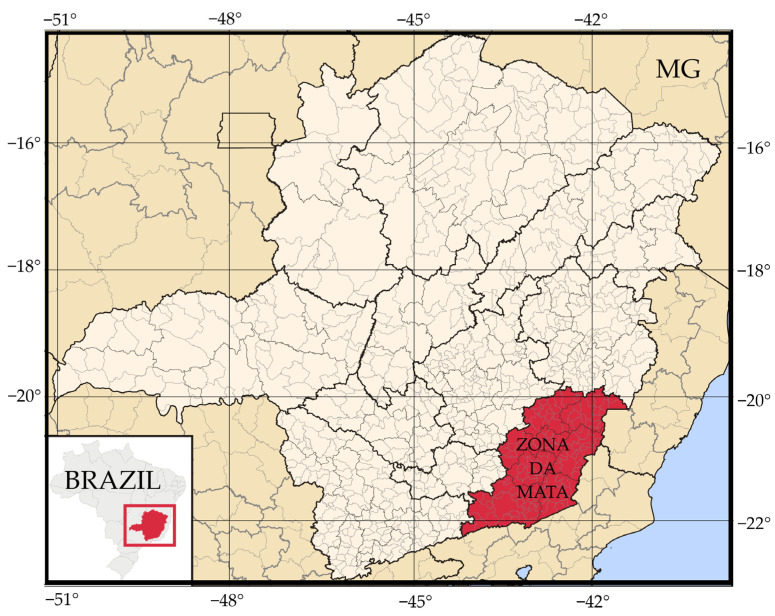
Map with the location of the state of Minas Gerais (MG) in Brazil and the Zona da Mata region (Latitude: between 20° and 22° South and Longitude: between 41° and 44° West). Adapted from Abreu (2006) [71].

**Figure 2 animals-13-00675-f002:**
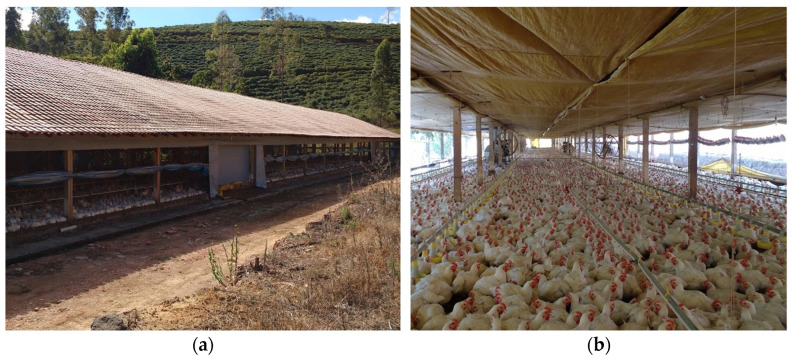
Open facilities for the production of broilers typical of the Zona da Mata of Minas Gerais (MG), Brazil; (**a**) external view and (**b**) internal view.

**Figure 3 animals-13-00675-f003:**
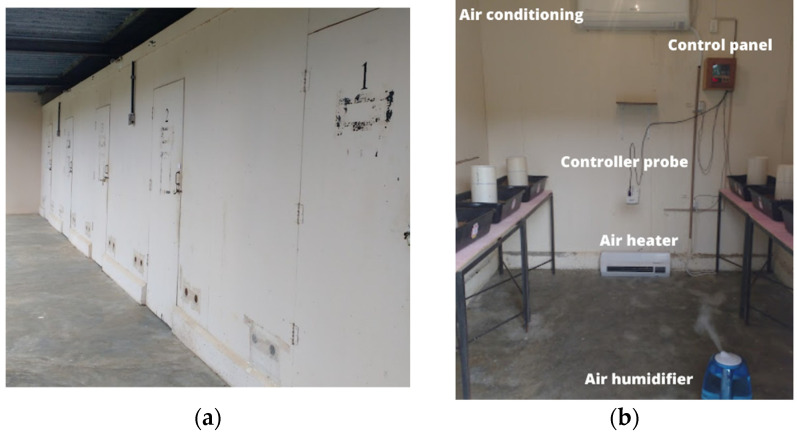
Images from climatic chambers; (**a**) external view and (**b**) internal view with the main equipment used.

**Figure 4 animals-13-00675-f004:**
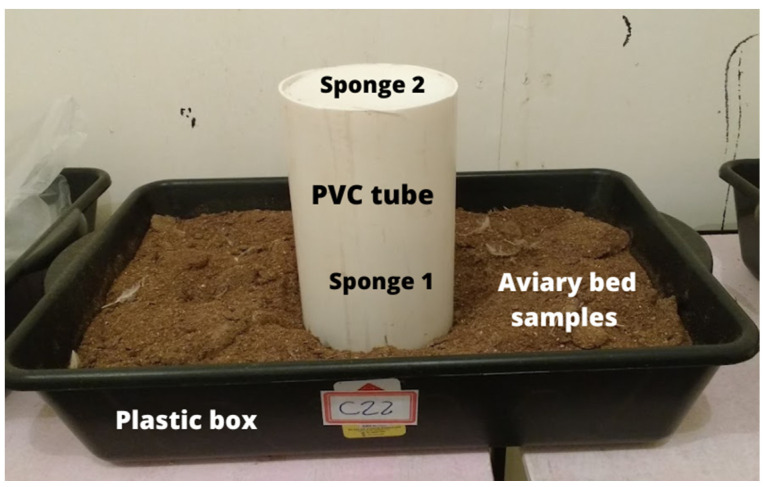
Image of the plastic box with the poultry litter sample and the volatilized ammonia capture device Saraz Method for Determination of Ammonia Emissions (SMDAE).

**Figure 5 animals-13-00675-f005:**
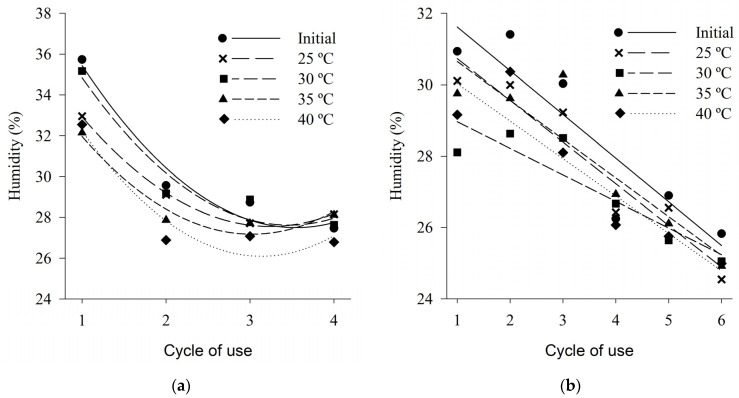
Moisture content of wood shaving bed (**a**) and coffee husk bed (**b**) samples in different cycles of use and heat treatments.

**Figure 6 animals-13-00675-f006:**
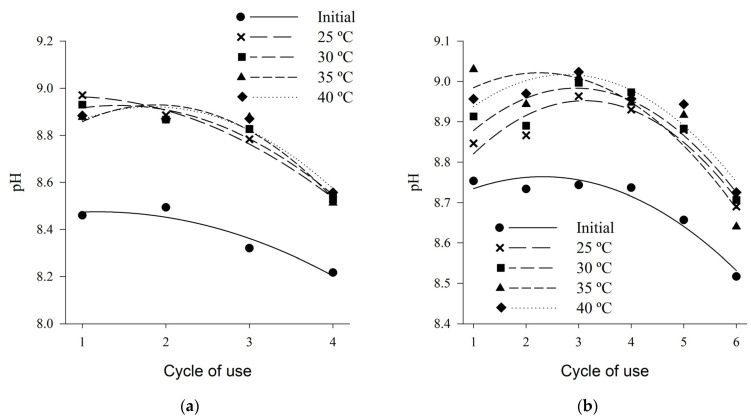
pH of wood shaving bed (**a**) and coffee husk bed (**b**) samples in different cycles of use and heat treatments.

**Figure 7 animals-13-00675-f007:**
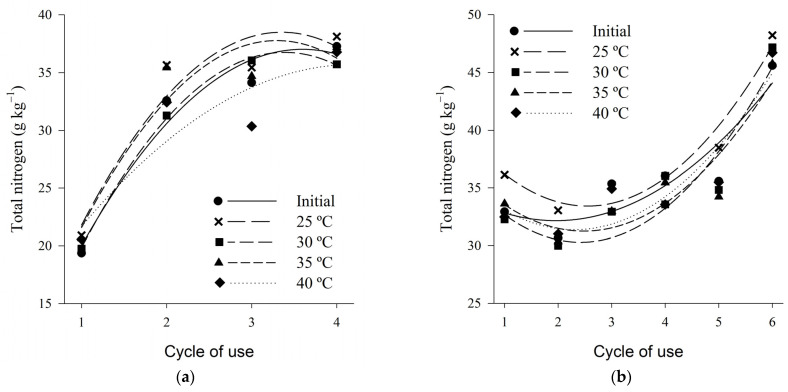
Total nitrogen content of wood shaving bed (**a**) and coffee husk bed (**b**) samples in different cycles of use and heat treatments.

**Figure 8 animals-13-00675-f008:**
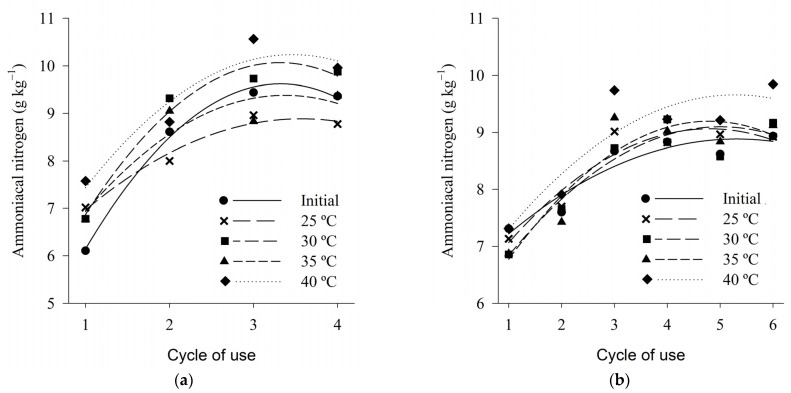
Ammonia nitrogen of wood shaving bed (**a**) and coffee husk bed (**b**) samples in different cycles of use and heat treatments.

**Figure 9 animals-13-00675-f009:**
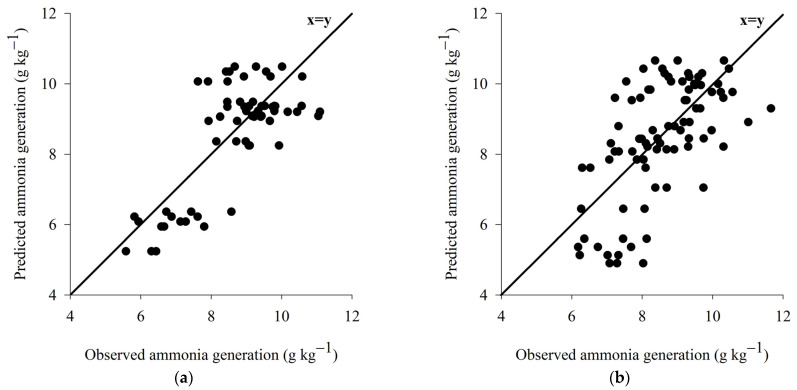
Representation of the prediction models of the ammonia generation potential for wood shaving bed (**a**) and coffee husk bed (**b**).

**Figure 10 animals-13-00675-f010:**
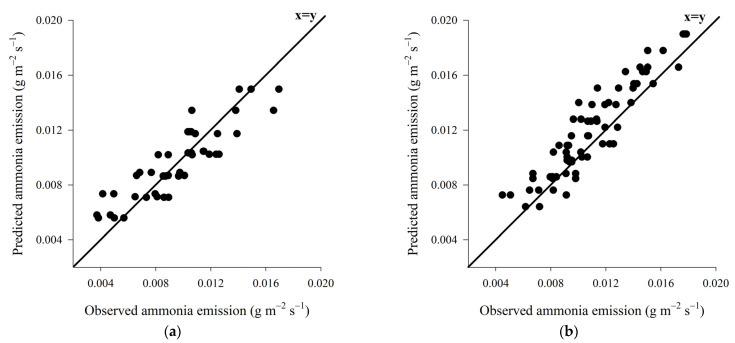
Representation of the ammonia emission prediction models for wood shaving bed (**a**) and coffee husk bed (**b**).

**Table 1 animals-13-00675-t001:** Adjusted equations of the moisture behavior of the wood shaving bed and coffee husk bed in relation to the cycle of use, in the different air temperatures.

Air Temperature	Equation	R^2^	SEE
Wood shaving bed			
Initial	U = 42.91 − 8.69 × C + 1.,22 × C^2^	0.96	1.29
25 °C	U = 38.81 − 6.95 × C + 1.07 × C^2^	0.99	0.14
30 °C	U = 41.89 − 8.24 × C + 1.19 × C^2^	0.93	1.48
35 °C	U = 37.88 − 7.08 × C + 1.17 × C^2^	0.95	0.79
40 °C	U = 39.32 − 8.43 × C + 1.34 × C^2^	0.91	1.41
Coffee husk bed			
Initial	U = 32.84 − 1.22 × C	0.83	1.14
25 °C	U = 31.90 − 1.17 × C	0.91	0.75
30 °C	U = 29.71 − 0.74 × C	0.82	0.73
35 °C	U = 31.74 − 1.09 × C	0.82	1.05
40 °C	U = 31.08 − 1.05 × C	0.85	0.92

U—humidity (%); C—cycle of use; R^2^—coefficient of determination; SEE—standard error of the estimate.

**Table 2 animals-13-00675-t002:** Equations adjusted for pH determination of wood shaving and coffee husk beds.

Air Temperature	Equation	R^2^	SEE
Wood shaving bed			
Initial	pH = 8.43 + 0.08 × C − 0.03 × C^2^	0.92	0.06
25 °C	pH = 8.93 + 0.07 × C − 0.04 × C^2^	0.99	0.03
30 °C	pH = 8.80 + 0.16 × C − 0.06 × C^2^	0.96	0.06
35 °C	pH = 8.61 + 0.34 × C − 0.08 × C^2^	0.92	0.08
40 °C	pH = 8.66 + 0.27 × C − 0.07 × C^2^	0.93	0.07
Coffee husk bed			
Initial	pH = 8.67 + 0.08 × C − 0.02 × C^2^	0.94	0.03
25 °C	pH = 8.67 + 0.18 × C − 0.03 × C^2^	0.89	0.03
30 °C	pH = 8.74 + 0.16 × C − 0.03 × C^2^	0.86	0.05
35 °C	pH = 8.90 + 0.11 × C − 0.02 × C^2^	0.73	0.07
40 °C	pH = 8.82 + 0.14 × C − 0.02 × C^2^	0.83	0.04

C—cycle of use; R^2^—coefficient of determination; SEE—standard error of the estimate.

**Table 3 animals-13-00675-t003:** Equations adjusted for the behavior of total bed nitrogen according to the cycle of use.

Air Temperature	Equation	R^2^	SEE
Wood shaving bed			
Initial	NT = 4.41 + 18.12 × C − 2.52 × C^2^	0.95	2.96
25 °C	NT = 4.59 + 20.22 × C − 3.01 × C^2^	0.91	3.98
30 °C	NT = 2.69 + 20.11 × C − 2.97 × C^2^	0.99	0.37
35 °C	NT = 4.39 + 20.26 × C − 3.07 × C^2^	0.90	4.20
40 °C	NT = 11.57 + 11.46 × C − 1.36 × C^2^	0.82	5.01
Coffee husk bed			
Initial	NT = 34.95 − 2.86 × C + 0.73 × C^2^	0.83	2.71
25 °C	NT = 40.97 − 5.92 × C + 1.16 × C^2^	0.91	1.61
30 °C	NT = 37.22 − 5.74 × C + 1.18 × C^2^	0.82	2.56
35 °C	NT = 37.69 − 5.19 × C + 1.05 × C^2^	0.74	2.73
40 °C	NT = 36.64 − 4.57 × C + 0.99 × C^2^	0.76	2.78

NT—total nitrogen (g·kg^−1^); C—cycle of use; R^2^—coefficient of determination; SEE—standard error of the estimate.

**Table 4 animals-13-00675-t004:** Adjusted equations of the ammoniacal nitrogen behavior of the beds according to their cycle of use.

Air Temperature	Equation	R^2^	SEE
Wood shaving bed			
Initial	NA = 2.50 + 4.28 × C − 0.64 × C^2^	0.99	0.17
25 °C	NA = 5.17 + 2.08 × C − 0.29 × C^2^	0.97	0.25
30 °C	NA = 3.51 + 3.96 × C − 0.59 × C^2^	0.97	0.41
35 °C	NA = 4.43 + 2.93 × C − 0.43 × C^2^	0.87	0.72
40 °C	NA = 4.69 + 3.20 × C − 0.46 × C^2^	0.92	0.63
Coffee husk bed			
Initial	NA = 6.36 + 0.95 × C − 0.09 × C^2^	0.89	0.29
25 °C	NA = 5.87 + 1.33 × C − 0.14 × C^2^	0.91	0.30
30 °C	NA = 5.56 + 1.41 × C − 0.14 × C^2^	0.89	0.39
35 °C	NA = 5.32 + 1.61 × C − 0.17 × C^2^	0.86	0.49
40 °C	NA = 6.07 + 1.36 × C − 0.13 × C^2^	0.81	0.57

NA—ammoniacal nitrogen (g·kg^−1^); C—cycle of use; R^2^—coefficient of determination; SEE—standard error of the estimate.

**Table 5 animals-13-00675-t005:** Adequacy measurements of the models for the prediction of ammonia generation in wood shavings bed and coffee husks bed, estimated by the cross-validation technique.

Items	Wood Shaving Bed	Coffee Husk Bed
QMEP Partition (%)		
Mean deviation	2.94	1.58
Systematic deviation	33.29	15.88
Random error	63.76	82.53
Ranging from 0 to 1		
CCC	0.722	0.497
ρ (precision)	0.747	0.514
C_b_ (accuracy)	0.970	0.978
R^2^	0.559	0.265

QMEP = Mean square of prediction error; CCC = Correlation coefficient and correlation; ρ = Estimated correlation coefficient; C_b_ = Deviation correction factor; R^2^ = Determination coefficient.

**Table 6 animals-13-00675-t006:** Measurements of adequacy of models for the prediction of ammonia emissions in wood shavings bed and coffee husks bed, estimated by cross-validation technique.

Items	Wood Shaving Bed	Coffee Husk Bed
QMEP Partition (%)		
Mean deviation	2.94	1.58
Systematic deviation	33.29	15.88
Random error	63.76	82.53
Ranging from 0 to 1		
CCC	0.722	0.497
ρ (precision)	0.747	0.514
C_b_ (accuracy)	0.970	0.978
R^2^	0.559	0.265

QMEP = Mean square of prediction error; CCC = Correlation coefficient and correlation; ρ = Estimated correlation coefficient; C_b_ = Deviation correction factor; R^2^ = Determination coefficient.

## Data Availability

The data presented in this study are available on request from the corresponding author.

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
