# Peer review of "Potential for Ammonia Generation and Emission in Broiler Production Facilities in Brazil"

_animals, 2023, doi:10.3390/ani13040675_

Round 1
Reviewer 1 Report
This paper proposed prediction models for predicting NH3 generation and emission potentials under a hot climatic condition. The proposed models could be useful in countries with the mentioned climatic conditions. However, I have several main concerns as below:
1) In the “Introduction” section, the authors didn’t give a comprehensive overview of the existing methods in predicting NH3 emissions from livestock and/or poultry farms. What’s the current status in related researches? The significance of the study needs to be further clarified.
2) The “Materials and Methods” section didn’t provide enough details of the methods adopted in this study. There is no any figure or photo showing the poultry facilities, the “4 (four) climatic chambers”, the aviary beds constituted by wood shavings and coffee husks, and the experimental design. As a study aiming to develop a prediction model for NH3 emissions, there is no any equations necessary to illustrate the calculation/estimation of NH3 emissions. The lacking of essential details in methodology spoils the robustness of this paper. Besides, the logical organization of paragraph structure in the method section needs to be strengthened.
3) It seems like the proposed prediction model for NH3 generation potential and emission potential estimation were obtained using SAS procedure REG. This method could be used to explore the relationships between dependents (NH3 generation/emission potentials) and independents (T and C). However, it became problematic if the they are compared (using scatter plots) to validate the accuracy of the prediction models (Figure 1 and 2) due to the problem of autocorrelation.
4) In the “Results and Discussions” section, essential observational data including moisture content, pH, total nitrogen, ammoniacal nitrogen, as well as the original NH3 generation and emission, were all NOT shown in the manuscript.
5) What’s the “potential generation”? Please give some explanation about this concept and its relationship with “potential emission”. Why is “potential generation” so important?
Other specific comments are:
Line 55-56: “GHG”(Green house gases) is a problem in the fields of ecology and climatology, but not an environmental problem. Whereas “Pollutants” and “Air quality” are concepts in the environmental field. Although ammonia has concurrently both environmental and greenhouse effects, the sentence (Line 55-56) has conceptually logic problem. The sentence in Line 49 has the similar problem. Please revise them accordingly.
Line 64: Through the research of this study, which factor could serve as such an indicator to relate the potential for ammonia generation and emissions with the animal production?
Line 68-70: Please give some of the names and corresponding references of the existing “methodologies available for quantification of ammonia”, and explain why these methodologies may not suitable for the case in the present study.
Line 70-71: What does “open facilities” look like? I suggest give a photo in the methods section.
Line 126-127: How was the NH3 flux measured? What’s the principle of measurement? What’s the instrument/devices used? What’s the uncertainty of measurement?
Line 144-146: The calculation or estimation of NH3 generation potential or emission potential doesn’t belong to the “experiment” part, but should be regarded as something like calculation (or prediction/estimation) categories.
Line 165: What does “cross-validation” mean? Could you give a brief description of it?
Line 198-201(Table 1): What does “C2” in the equations mean?
Table 1 to Table 4: The results may also be presented in figures so that they look more intuitive to readers.
Author Response
Reviewer: This paper proposed prediction models for predicting NH3 generation and emission potentials under a hot climatic condition. The proposed models could be useful in countries with the mentioned climatic conditions. However, I have several main concerns as below:
Authors: Dear reviewer, we thank you for your comments, that were very important for improving the manuscript quality. We have revised the manuscript addressing your concerns in a point-by-point response that are highlighted in red the revised manuscript.
Reviewer: 1) In the “Introduction” section, the authors didn’t give a comprehensive overview of the existing methods in predicting NH3 emissions from livestock and/or poultry farms. What’s the current status in related researches? The significance of the study needs to be further clarified.
Authors: We appreciate your observation and therefore the text of the Introduction has been supplemented with the inclusion of sentences, paragraphs and new citations to address the issues mentioned. Please see lines 48-74 and 89-106 in the revised manuscript.
Reviewer: 2) The “Materials and Methods” section didn’t provide enough details of the methods adopted in this study. There is no any figure or photo showing the poultry facilities, the “4 (four) climatic chambers”, the aviary beds constituted by wood shavings and coffee husks, and the experimental design. As a study aiming to develop a prediction model for NH3 emissions, there is no any equations necessary to illustrate the calculation/estimation of NH3 emissions. The lacking of essential details in methodology spoils the robustness of this paper. Besides, the logical organization of paragraph structure in the method section needs to be strengthened.
Authors: Thanks. We hope to have met all of your notes in the modifications that we have made to practically all of the text in the Material and Methods section. We included more detailed descriptions, added images and equations to improve the methodology, convey greater reliability in the research and seek greater robustness in the manuscript. Please see lines 124-229 in the revised manuscript.
Reviewer: 3) It seems like the proposed prediction model for NH3 generation potential and emission potential estimation were obtained using SAS procedure REG. This method could be used to explore the relationships between dependents (NH3 generation/emission potentials) and independents (T and C). However, it became problematic if the they are compared (using scatter plots) to validate the accuracy of the prediction models (Figure 1 and 2) due to the problem of autocorrelation.
Authors: Dear reviewer, our intention here was to illustrate that the predicted values made by our models were quite like the observed values. We understand and agree with you that autocorrelation may be a problem. Therefore, we used cross-validation to test the accuracy and precision of our models, because this is a robust technique, which has no problems regarding autocorrelation.
Reviewer: 4) In the “Results and Discussions” section, essential observational data including moisture content, pH, total nitrogen, ammoniacal nitrogen, as well as the original NH3 generation and emission, were all NOT shown in the manuscript.
Authors: Thanks for the observation. In view of your comment, we decided to include the graphs with the results of the studied variables, which contain the initial (original) values and the values after the treatments. Please see lines 275-285, 316-322, 348-355 and 372-380 in the revised manuscript.
Reviewer: 5) What’s the “potential generation”? Please give some explanation about this concept and its relationship with “potential emission”. Why is “potential generation” so important?
Authors: The potential for ammonia generation represents the maximum amount of ammonia that can be generated in the bed given possible variations in ambient air temperature. Depending on the maximum ammonia that can be generated (generation potential) in the beds of the facilities, one can have an expectation of the maximum ammonia that can be emitted (emission potential), since the emission depends on many other variables but it is also limited by the ammonia content of the bed. While the potential for ammonia emissions is important for environmental and ecological issues, the potential for ammonia generation is important because of local problems related to the concentration of ammonia inside the installation, mainly because of the damage to the health of animals and workers, which will be a function of the ammonia concentration and the time of exposure to ammonia.
Thanks for the questions. Part of that answer was included in the introduction. Please see lines 82-85 in the revised manuscript.
Reviewer: Other specific comments are:
Reviewer: Line 55-56: “GHG”(Green house gases) is a problem in the fields of ecology and climatology, but not an environmental problem. Whereas “Pollutants” and “Air quality” are concepts in the environmental field. Although ammonia has concurrently both environmental and greenhouse effects, the sentence (Line 55-56) has conceptually logic problem. The sentence in Line 49 has the similar problem. Please revise them accordingly.
Authors: Thanks for the observation. There really were conceptual mistakes in that paragraph so this whole paragraph was restructured. Please see lines 48-74 in the revised manuscript.
Reviewer: Line 64: Through the research of this study, which factor could serve as such an indicator to relate the potential for ammonia generation and emissions with the animal production?
Authors: We referred to the variables of the thermal environment and management, such as the variables we used in the present study, which are air temperature and bed reuse cycle. We modified the text to “there isn’t an indicator relating the potential for ammonia generation and emissions with variables that are easier to measure and/or obtain in animal production facilities such as characteristics of the thermal environment (air temperature) and facility management (bed type and bed reuse cycle). Please see lines 82-85 in the revised manuscript.
Reviewer: Line 68-70: Please give some of the names and corresponding references of the existing “methodologies available for quantification of ammonia”, and explain why these methodologies may not suitable for the case in the present study.
Authors: Thanks for the suggestion. We added a few paragraphs in the introduction including your suggestions. Please see lines 89-106 in the revised manuscript.
Reviewer: Line 70-71: What does “open facilities” look like? I suggest give a photo in the methods section.
Authors: We modified the text and included a general description of the open facilities ….“without side walls in the facilities, using only a screen, which makes the internal environment subject to external variations, in addition to making it impossible to control the volume of internal air and making it difficult to define air inlets and outlets in the facilities”… (Please see lines 109-113 in the revised manuscript) and as suggested we also added in the Material and Methods item images with external and internal views of a of the installations that were included in the experiment (Please see lines 142-152 in the revised manuscript).
Reviewer: Line 126-127: How was the NH3 flux measured? What’s the principle of measurement? What’s the instrument/devices used? What’s the uncertainty of measurement?
Authors: Thanks! In fact, this information was vague in the text because we only cited the original article of the SMDAE methodology (Osorio-Saraz et al., 2014) but we decided to include more information that addresses all your questions in addition to an image with an assembly scheme of the equipment used in this research and also included the equation used to determine the ammonia flow. Please see along Material and Methods, in item 2.2 (Please see lines 183-229 in the revised manuscript).
Reviewer: Line 144-146: The calculation or estimation of NH3 generation potential or emission potential doesn’t belong to the “experiment” part, but should be regarded as something like calculation (or prediction/estimation) categories.
Authors: Sorry if we didn't quite understand what you meant, but for what we understand, we decided to include the equations used to determine the ammoniacal nitrogen content, which was used to estimate the potential for ammonia generation, and the equation for determining the ammonia emission flow, used to estimate the ammonia emission potential. Please see lines 216-229 in the revised manuscript.
Reviewer: Line 165: What does “cross-validation” mean? Could you give a brief description of it?
Authors: As described in the main text, cross-validation is a stats technique used to assess the precision and accuracy of predictive models. The cross-validation uses the same dataset, which was used to develop the models, to test them. Thus, there is no problem regarding datasets with different variances. Briefly, the cross-validation technique consists of splitting the dataset into two parts, the first one is used to fit the models and the second to test the equations. After this process stats of good-of-fit are accessed, such as R², RMSE, CCC, etc, this process is repeated 1000 times and the average of these good-of-fit stats are reported and used to classify the models, regarding its accuracy and precision.
More information regarding cross-validation it is available in the following references:
Efron, B., 1983. Estimating the error rate of a prediction rule: improvement on cross-validation. Journal of American Statistical Association 78, 316–331.
Tedeschi, L.O. 2006. Assessment of the adequacy of mathematical models. Agric. Syst. 89:225–247. doi:10.1016/j.agsy.2005.11.004.
McMeniman JP, Tedeschi LO, Defoor PJ, Galyean ML. Development and evaluation of feeding-period average dry matter intake prediction equations from a commercial feedlot database. J Anim Sci. 2010;88: 3009–3017. doi:10.2527/jas.2009-2626
Davison, A.C., D. V. Hinkley, and G. A. Young. 2003. Recent Developments in Bootstrap Methodology. Stat. Sci. 18:141–157. doi:10.1214/ss/1063994969.
Reviewer: Line 198-201(Table 1): What does “C2” in the equations mean?
Authors: Thanks for the observation. Sorry, we made a typo in the table 1, the correct is C2 not C2 and corrected it in the manuscript text. The C means cycle of use and the 2 refers to the exponent that indicates that it is cycle raised to two or cycle squared. Please see Table 1 in the revised manuscript.
Reviewer: Table 1 to Table 4: The results may also be presented in figures so that they look more intuitive to readers.
Authors: Thanks for the suggestion. Initially, we thought of presenting the developed models together with the graphics that were included. However, we chose to continue presenting the models separately in a table because the inclusion of the equations together with the curves of the graphs generated excess text in the figure and compromised its quality. It is worth mentioning that the prominence of the models in the manuscript is important since prediction models are the main objective of the work.

Reviewer 2 Report
Dear Authors,
I consider that it is a interesting work. In my opinion paper is relative well structured and clarifying with respect to the object analyzed. Though the manuscript is well written and has scientific benefits, still some clarifications and improvements in the introduction and methodology part are required. Detailed comments in the attached file.

Author Response
Reviewer: Dear Authors, I consider that it is a interesting work. In my opinion paper is relative well-structured and clarifying with respect to the object analyzed. Though the manuscript is well written and has scientific benefits, still some clarifications and improvements in the introduction and methodology part are required. Detailed comments in the attached file.
Authors: Dear reviewer, we would like to thank you for your complement to our paper and for the opportunity to revise it. We have revised the manuscript considering all comments, and all concerns were addressed in a point-by-point response. Concerning your specific comments, they are addressed below, and changes are highlighted in red in the text.
Specific Comments:
Reviewer: Lines 49-50
This is true, but ammonia is not a greenhouse gas. It lacks a better logical connection to the presented research topic.
From acientific point of view, ammonia is formed from the breakdown of urea, which is found in large quantities in animal feces. Cellular urease is the enzyme-mediator of this process and the main source of fecal bacteria. Urea breaks down into ammonia and carbon dioxide.
Author: Thanks for that observation. The information was really confusing in the text and there were conceptual errors. All of your observations are pertinent and were all addressed in this paragraph, which has been completely reformulated. Please see lines 48-74 in the revised manuscript.
Reviewer: Lines 52-53
What is ideal air? Both in the building and on animal farms it is difficult to have air of satisfactory quality. This requires defining, removing the concept or developing the idea.
Authors: Very well remembered, the idea was actually very vague in the text and in order not to get too far out of focus in the manuscript, we decided to remove this point from the text.
Reviewer: Lines 52-53
There is a lack of general explanation what pollutants are mainly emitted in broiler production and why ammonia plays an important role in these pollutants.
Authors: Thank you for remembering this matter. The main pollutants have been included in the text in the new paragraph. Please see lines 53-62 in the revised manuscript.
Reviewer: Lines 54-55
What do they affect and what type of diseases are they? This needs clarification and possibly a literature reference. I suggest add a few positions of literature.
Ayman A. Swelum, Mohamed T. El-Saadony, Mohamed E. Abd El-Hack, Mahmoud M. Abo Ghanima, Mustafa Shukry, Rashed A. Alhotan, Elsayed O.S. Hussein, Gamaleldin M. Suliman, Hani Ba-Awadh, Aiman A. Ammari, Ayman E. Taha, Khaled A. El-Tarabily, (2021). Ammonia emissions in poultry houses and microbial nitrification as a promising reduction strategy. Science of The Total Environment, 781, 146978,
Herbut, P., Angrecka, S. (2014). Ammonia concentrations in a free-stall dairy barn. Annals Animal Science, 14(1), 153–166.
Oliveira, M.D.; Sousa, F.C.; Saraz, J.O.; Calderano, A.A.; Tinôco, I.F.F.; Carneiro, A.P.S. (2021). Ammonia Emission in Poultry Facilities: A Review for Tropical Climate Areas. Atmosphere 2021, 12, 1091.
Authors: The text has been modified and information added. All suggested references have also been included in the text. Please see lines 63-74 in the revised manuscript.
Reviewer: Line 56
Because ammonia is not a greenhouse gas it would be more beneficial to focus on the health and odor aspect.
Authors: Accordingly, this paragraph was completely modified and a new paragraph was created focusing on damage to health. Please see lines 63-74 in the revised manuscript.
Reviewer: Line 75
Rather, in this case, the generation of ammonia, since emissions will depend on further environmental factors.
Authors: Accordingly. Thanks for the observation. The text has been modified. Please see line 117 in the revised manuscript.
Reviewer: Line 76-77
However, I suggest limiting ourselves to the continent as South America, since such strategies exist in Europe.
Authors: According to your notation. The text has been modified. Please see line 118 in the revised manuscript.
Reviewer: Line 81
Are you sure about the emissions? Was it adequately addressed in the article?
Authors: Thanks for the question, but we believe so, for ammonia emissions as well. We developed models to estimate ammonia emissions as a function of air temperature (T) and the bed reuse cycle (C) for two types of bed (wood shavings and coffee husks) as presented in the last topic of the results (3.3 Ammonia emission potential). Please see line 439-472 in the revised manuscript.
Reviewer: Line 94
Describe the climatic conditions in general.
Authors: The general climatic conditions of the region have been added in the text. Please see line 134-138 in the manuscript in the revised manuscript.
Reviewer: Line 94-95
Maybe add a map with the location of the objects if that would not be a problem?
Authors: Thanks for the suggestion. Added Figure 1 with the map indicating the state of Minas Gerais (MG) in Brazil and the Zona da Mata region in the state of MG.
Reviewer: Lines 105-106
feeding and watering systems (lines)
Authors: The text has been modified. Please see line 162-163 in the revised manuscript.
Reviewer: Line 106
Check the correctness of the sentence. The samples were taken with a shovel. Right?
Authors: Yes. The text has been modified as suggested. The citation on the recommendation of the poultry litter sampling process that was used in this research was included. Please see line 163 in the revised manuscript.
Reviewer: Line 112
If they were typical, state the country of manufacture, company, model.
Authors: Climatic chambers were built in the research area. This information was included and Figure 3 was added with internal and external images of the climatic chambers. Please see lines 169-182 in the revised manuscript.
Reviewer: Line 126
Give the name, country of manufacture, company and model.
Authors: The equipment used to determine the ammonia emission flow was based on the SMDAE methodology by Osorio-Saraz et al. (2014). Therefore, it was assembled by the researchers themselves according to the cited author's recommendations, so we do not have the information you asked to include. But to make the text clearer, we have included a more detailed description of the method and also illustrative images of the experiment and the equipment used in this research. Please see line 183-198 in the revised manuscript.
Reviewer: Line 189
Uzyj lepszego slowa. To nie jest dobre.
Authors: Thanks for the observation. We change the term to “pattern”. Please see line in the revised manuscript. Please see line 272 in the revised manuscript.
Reviewer: Line 199
rather Air temperature
em vez de temperatura do ar
Authors: Modified. Please see Table 1 in the revised manuscript.
Reviewer: Line 227
Air temperature
Authors: Modified. Please see Table 2 in the revised manuscript.
Reviewer: Line 256
Air temperature
Authors: Modified. Please see Table 3 in the revised manuscript.
Reviewer: Line 272
Air temperature
Authors: Modified. Please see Table 4 in the revised manuscript.
Reviewer: Lines 361-376
It would be beneficial to introduce limitations to the studies presented, such as the area of application of the model due to the type of substrate used, No varying values of ventilation at the bedding, which affects the generation of ammonia, etc.
Authors: Thanks for the suggestion. The information has been included in the Conclusion. Please see lines 489-493 in the revised manuscript.
Reviewer: Line 376
I suggest adding a sentence regarding the Need of further research in real production conditions.
Authors: Thanks for the suggestion. The information has been included in the Conclusion. Please see lines 493-495 in the revised manuscript.

Round 2
Reviewer 1 Report
Thank you for the efforts the authors made to the revision of the manuscript. The revised version looks much improved and I think it already meets the requirements to be published in the Animals journal.
1. Please include the longitude and latitude information in Figure 1.
2. Line 222: Please check if there is a typo in this line.
Author Response
Reviewer 1 - Round 2
Reviewer: 1. Please include the longitude and latitude information in Figure 1.
Author: Dear reviewer, thank you for the suggestion. We include coordinates both on the map and in the figure description. Please see lines 139-142 in the revised manuscript.
Reviewer: 2. Line 222: Please check if there is a typo in this line.
Author: I'm sorry for the mistake. We corrected it by including the meaning of the pH that was missing. See line 223 in the revised manuscript.
